# Development of a Polygenic Risk Score for BMI to Assess the Genetic Susceptibility to Obesity and Related Diseases in the Korean Population

**DOI:** 10.3390/ijms241411560

**Published:** 2023-07-17

**Authors:** Nara Yoon, Yoon Shin Cho

**Affiliations:** Department of Biomedical Science, Hallym University, Chuncheon 24252, Republic of Korea; hidy2933@naver.com

**Keywords:** obesity, body mass index, polygenic risk score, single-nucleotide polymorphism, genome-wide association study, disease risk assessment

## Abstract

Hundreds of genetic variants for body mass index (BMI) have been identified from numerous genome-wide association studies (GWAS) in different ethnicities. In this study, we aimed to develop a polygenic risk score (PRS) for BMI for predicting susceptibility to obesity and related traits in the Korean population. For this purpose, we obtained base data resulting from a GWAS on BMI using 57,110 HEXA study subjects from the Korean Genome and Epidemiology Study (KoGES). Subsequently, we calculated PRSs in 13,504 target subjects from the KARE and CAVAS studies of KoGES using the PRSice-2 software. The best-fit PRS for BMI (PRS_BMI_) comprising 53,341 SNPs was selected at a *p*-value threshold of 0.064, at which the model fit had the greatest *R*^2^ score. The PRS_BMI_ was tested for its association with obesity-related quantitative traits and diseases in the target dataset. Linear regression analyses demonstrated significant associations of PRS_BMI_ with BMI, blood pressure, and lipid traits. Logistic regression analyses revealed significant associations of PRS_BMI_ with obesity, hypertension, and hypo-HDL cholesterolemia. We observed about 2-fold, 1.1-fold, and 1.2-fold risk for obesity, hypertension, and hypo-HDL cholesterolemia, respectively, in the highest-risk group in comparison to the lowest-risk group of PRS_BMI_ in the test population. We further detected approximately 26.0%, 2.8%, and 3.9% differences in prevalence between the highest and lowest risk groups for obesity, hypertension, and hypo-HDL cholesterolemia, respectively. To predict the incidence of obesity and related diseases, we applied PRS_BMI_ to the 16-year follow-up data of the KARE study. Kaplan–Meier survival analysis showed that the higher the PRS_BMI_, the higher the incidence of dyslipidemia and hypo-HDL cholesterolemia. Taken together, this study demonstrated that a PRS developed for BMI may be a valuable indicator to assess the risk of obesity and related diseases in the Korean population.

## 1. Introduction

Obesity is a medical condition involving the excessive accumulation of fat, which causes various health problems. As obesity can impair quality of life as a risk factor for numerous metabolic diseases and cancer [1], the recent increase in its prevalence poses a threat to wellbeing in human populations [2]. A number of genetic studies have been conducted to understand the genetic basis of obesity, which is known as a heritable trait [3,4,5,6,7,8]. To date, genome-wide association studies (GWASs) have identified over 250 common genetic variants for body mass index (BMI) [9,10], a simple index generally used as an indicator of obesity [11].

Most obesity-related genes are involved in appetite-related signals, adipocyte growth and differentiation, energy expenditure regulation, or insulin metabolism and adipose tissue inflammation [12]. As an example, *LEP* encodes leptin that is an adipocyte-secreted hormone involved in appetite-related signals [13]. The circulating levels of leptin are known to correlate closely with overall adiposity. Several common variants (located in/near *LEP*, *SLC32A1*, *GCKR*, *CCNL1*, and *FTO*) influencing circulating leptin levels have been identified by a GWAS in individuals of European ancestry [14].

There is insufficient biological evidence that many of the loci identified from numerous GWASs for obesity play a causal role by directly promoting or preventing weight gain. This uncertainty has been a major barrier to treating obesity with the power of demographic genomics. However, GWAS variants may be useful in predicting individual susceptibility to disease by developing risk assessment models.

As is known, complex traits such as obesity are influenced by multiple genes, each with a small effect [15]. Furthermore, most genetic variants identified by GWASs usually account for only small effect sizes for a trait [16]. Therefore, the approach of applying a single variant has limitations in predicting complex traits. Since the heritability of many complex traits is determined among many variants with small effect sizes, it has been proposed that more accurate prediction can be achieved using genome-wide variants instead of several significantly related variants [17].

The polygenic risk score (PRS), a weighted sum of the number of risk alleles carried by an individual, has recently attracted attention as it has the potential to evaluate the explanatory power of polygenes and predict the risk of common diseases in a population [18,19]. The availability of large datasets from large-scale GWASs and the advance of computational methods to calculate PRSs have facilitated the application of polygenic risk profiling to identify groups of individuals susceptible to disease [20]. Indeed, PRS analysis has been conducted for several common diseases, including coronary artery disease, atrial fibrillation, type 2 diabetes, diabetic retinopathy, inflammatory bowel disease, dyslipidemia, breast cancer, and colorectal cancer [20,21,22,23].

Software programs capable of processing large amounts of data have been developed to calculate PRS, including LDpred (v.1.4.7) [24], lassosum (v.0.4.4) [25], and PRSice-2 (v.2.3.5) [26]. In this study, we aimed to develop a genome-wide PRS for obesity in the Korean population using PRSice-2, which has a short running time and small memory occupancy, regardless of the sample size, and has predictive power equivalent to that of LDpred and lassosum [26]. We also further extended the PRS derived in this study to predict the incidence of obesity and related diseases such as hypertension, dyslipidemia, and type 2 diabetes in the Korean population.

## 2. Results

### 2.1. Production of Base Data for Computing BMI PRS

Summary statistics needed as base data to calculate the BMI PRS were obtained from a GWAS for BMI using 8,056,211 variants of 57,110 HEXA cohort subjects (Table 1). Association analysis between SNPs and BMI revealed 20 independent SNPs reaching genome-wide significance (*p*-value < 5 × 10^−8^) (Appendix A). With the exception of rs143349795, most SNPs with genome-wide significance detected in this study have been identified in East Asian GWASs for BMI [8]. The SNP rs143349795 is located in the intron of *Cyclic Nucleotide Binding Domain Containing 2* (*CNBD2*) (Appendix A). The protein encoded by *CNBD2* possessing cAMP-binding activity is known to be involved in spermatogenesis [27]. The above findings suggest the value of also elucidating the functional role of CNBD2 in obesity.

### 2.2. Derivation of PRS for BMI

We computed the PRS for BMI using the GWAS summary statistics of 57,110 HEXA cohort subjects as base data and 13,504 genotype data from KARE and CAVAS cohorts as target data (Figure 1). Using PRSice-2 software, the best-fit PRS for BMI (PRS_BMI_) was selected at a *p*-value threshold of 0.064, at which the model fit had the greatest *R*^2^ score (Figure 2).

### 2.3. Validation of PRS_BMI_ for Obesity-Related Quantitative Traits/Diseases

The derived PRS_BMI_ was tested for its association with obesity-related quantitative traits by correlation and linear regression analyses (Table 2). Linear regression analysis demonstrated that PRS_BMI_ was most strongly associated with BMI (*p* = 1.36 × 10^−73^). Indeed, the BMI measurements were shown to differ significantly among the quartile groups of PRS_BMI_ (Figure 3). The proportion of the variance of BMI explained by PRS_BMI_ was 2.4%. PRS_BMI_ was also found to be significantly associated with several obesity-related quantitative traits including systolic blood pressure (SBP) (*p* = 2.63 × 10^−6^), diastolic blood pressure (DBP) (*p* = 2.90 × 10^−5^), fasting insulin (INS0) (*p* = 8.15 × 10^−3^), high-density lipoprotein cholesterol (HDLC) (*p* = 9.07 × 10^−3^), and triglyceride (TG) (*p* = 2.50 × 10^−2^) (Table 2).

Logistic regression was performed to demonstrate the association between PRS_BMI_ and obesity-related diseases. Significant associations of PRS_BMI_ with obesity (*p* = 8.73 × 10^−45^), hypertension (*p* = 6.84 × 10^−4^), and hypo-HDL cholesterolemia (*p* = 2.75 × 10^−2^) were detected in subjects of the target dataset (Table 3). It was estimated that the highest-risk group of PRS_BMI_ (the fourth quartile, Q4) had approximately 2-fold, 1.1-fold, and 1.2-fold risk for obesity, hypertension, and hypo-HDL cholesterolemia, respectively, in comparison to the lowest-risk group (the first quartile, Q1) in the test population of the target dataset (Figure 4).

### 2.4. Prevalence of Obesity and Related Diseases among Genetic Risk Groups in the Population

The prevalence of obesity and related diseases (such as hypertension, T2D, dyslipidemia, hypo-HDL cholesterolemia, hyper-LDL cholesterolemia, and hyperglyceridemia) was compared according to each decile group of PRS_BMI_ in about 13,000 subjects in the target dataset (from KARE and RURAL cohorts). Significant correlations were detected between the decile groups of PRS_BMI_ and the prevalence of obesity, hypertension, and hypo-HDL cholesterolemia. In the test population, there were differences in the prevalence of obesity, hypertension, and hypo-HDL cholesterolemia of about 26%, 2.8%, and 3.9% between the highest- and lowest-risk groups, respectively (Figure 5).

### 2.5. Incidence of Obesity and Related Diseases among Genetic Risk Groups in the Population

We evaluated whether the PRS_BMI_ that we developed predicts the incidence of obesity and related diseases. It is assumed that the higher the PRS, the higher the incidence of such conditions. In this study, we divided the PRS_BMI_ into quartiles and analyzed the predictive power of the quartile groups for the incidence of diseases. Follow-up survey data from 2001 to 2016 from the KARE cohort subjects were used for this purpose. Here, individuals who never been followed up were excluded from the analysis.

Kaplan–Meier survival analysis followed by a log-rank test demonstrated that the incidences of dyslipidemia and hypo-HDL cholesterolemia differed significantly among the quartile groups of PRS_BMI_, while other diseases did not show significant differences (Figure 6). For dyslipidemia and hypo-HDL cholesterolemia, the higher-risk group of PRS_BMI_ had a higher incidence of diseases in the test population.

## 3. Discussion

Genome-wide association studies (GWASs) have identified over 55,000 unique loci for numerous common diseases and traits since the first GWAS was reported in 2005 [28]. To date, more than 70,000 common SNPs with genome-wide significance (association *p*-value ≤ 5.0 × 10^−8^) have been accumulated and are publicly available in the GWAS catalog (https://www.ebi.ac.uk/gwas/, accessed on 1 May 2023). The majority of GWASs have been conducted in populations of European ancestry, with only about 10% of all GWAS subjects being of non-European descent [28]. For example, in research on studies reported from 2005 to 2016, East Asian participants accounted for only 9% of the ancestral data included in the GWAS catalog (https://www.ebi.ac.uk/gwas/, accessed on 1 May 2023). This disproportional representation of ancestry populations prevents an accurate understanding of the transferability of GWAS results across populations and makes it difficult to apply results informed by genetic research to clinical care.

Like many other complex traits, obesity has been a major trait subjected to large-scale GWA analysis. To date, GWASs have detected over 250 common genetic variants for BMI [9,10], including those from East Asian populations [3,5,8]. In this study, we performed a GWA analysis for BMI in the Korean population in the part of the generation of base data for the PRS calculation. The GWA analysis using 57,110 HEXA cohort subjects detected 20 SNPs showing genome-wide significant associations with BMI. Of these, variants in or near *FTO* (*P*_GWAS_ = 9.8 × 10^−24^), *SEC16B* (*P*_GWAS_ = 1.4 × 10^−26^), *BDNF* (*P*_GWAS_ = 2.7 × 10^−21^), and *TMEM18* (*P*_GWAS_ = 7.4 × 10^−12^) had also been discovered in previous studies [29,30,31]. Meanwhile, the SNP rs143349795 in *CNBD2* (*P*_GWAS_ = 1.2 × 10^−8^) was detected for the first time in this study. The fact that rs143349795 is monomorphic in populations of European ancestry explains why no association of this SNP with BMI has been detected in Europeans.

As most of the SNPs identified in GWASs are in introns and intergenic regions, it is believed that they exert small effects on disease risk and explain only a fraction of the heritability. As such, loci identified in GWASs may not make major contributions to disease prediction or causality. To overcome this limitation, a PRS combining risk alleles across the whole genome has been developed to improve the prediction of target diseases or traits [32]. With summary statistics for most GWASs being publicly available, the Polygenic Score (PGS) catalog has recently been established to provide information on PRSs to predict the genetic predisposition to diverse phenotypes such as diseases (https://www.pgscatalog.org/). As is typically the case for GWASs, most PRSs have been developed in populations of European ancestry. This European bias presents a crucial limitation in predicting the risk of diseases across populations globally.

Against this background, we developed a PRS for predicting obesity in the Korean population. The best-fit PRS generated from our GWAS for BMI (PRS_BMI_) showed strong associations with BMI (*p* = 1.36 × 10^−73^) and obesity (*p* = 8.73 × 10^−45^) from linear regression and logistic regression analyses, respectively. The proportion of variance in BMI explained by PRS_BMI_ was about 2.4% in the Korean population. Of several obesity-related quantitative traits, SBP, DBP, INS0, HDL, and TG also showed significant associations with PRS_BMI_ (Table 2). These results match the significant associations of PRS_BMI_ with obesity-related diseases such as hypertension (*p* = 6.84 × 10^−4^) and hypo-HDL cholesterolemia (*p* = 2.75 × 10^−2^) well (Table 3). Based on these results, we further aimed to examine whether PRS_BMI_ could reliably predict the prevalence of obesity and related diseases in the Korean population. The distribution of PRS_BMI_ demonstrated that individuals with a high PRS_BMI_ tend to be more susceptible to obesity than those with a low PRS_BMI_. In addition, we observed that the prevalence of obesity-related diseases such as hypertension and hypo-HDL cholesterolemia increased in the high-risk PRS_BMI_ group.

In an effort to predict the incidence of obesity and related diseases using PRS_BMI_ in the Korean population, we also performed Kaplan–Meier survival analysis using follow-up survey data from 2001 to 2016 from the KARE cohort, one of our study cohorts. Kaplan–Meier survival analysis followed by a log-rank test demonstrated that individuals in the high-risk PRS_BMI_ group a had nominally significant higher likelihood of developing dyslipidemia and hypo-HDL cholesterolemia. Meanwhile, no clear increases in the incidences of other diseases were observed in the high-risk PRS_BMI_ group in this study. This result may be partly due to the small sample size in the follow-up longitudinal study of the KARE cohort. To generalize prediction of the incidence of diseases using PRS_BMI_ in the Korean population, it may be necessary to increase the sample size.

Given the need for studies aimed at developing genome-wide PRS in more diverse populations, it is meaningful that our study is, to the best of our knowledge, the first to develop and test a genome-wide PRS for obesity and related diseases in an East Asian population. Our results demonstrated the promise of the PRS_BMI_ developed in this study as a useful index to predict obesity and related diseases in the Korean population. Accordingly, it was suggested that PRS_BMI_ could be used clinically to prevent obesity and related diseases in advance. Subsequent large-scale studies of PRSs for diverse phenotypes such as diseases may open up various avenues for the application of genetic findings in a clinical context.

## 4. Materials and Methods

### 4.1. Study Subjects

Subjects for the association analyses were recruited from the Korean Genome and Epidemiology Study (KoGES). KoGES is a consortium project designed by the Korea Disease Control and Prevention Agency and consists of population-based and gene–environmental study cohorts comprising approximately 225,000 participants [33]. We used the epidemiological data from three population-based cohorts in KoGES: the RURAL cohort derived from the KoGES cardiovascular disease association study (CAVAS), the KoGES Ansan and Ansung study cohort (KARE), and the KoGES Health Examinees study cohort (HEXA) [34,35,36] (Table 1). The subjects of the KARE cohort designed for longitudinal prospective study have been examined every 2 years since 2001 [33].

### 4.2. Genotyping, Quality Control, and Imputation

Approximately 71,000 individuals from three population-based cohorts of KoGES were genotyped with the Korea Biobank Array (KBA) chip [37]. As quality control (QC), samples with sample call rate < 97%, excessive heterozygosity, excessive singletons, gender discrepancy, and cryptic first-degree relatives were removed. SNPs with SNP call rate < 95%, minor allele frequency (MAF) < 1%, and Hardy–Weinberg equilibrium (HWE) *p*-value < 1 × 10^−6^ were excluded from subsequent analyses.

After phasing genotype data using Eagle v2.3, SNP imputation was performed with IMPUTE4 using 1000 Genomes Project phase 3 and Korean reference genome (397 samples) as a reference panel. After imputation, SNPs with INFO score < 0.8 and MAF < 1% were eliminated.

### 4.3. Phenotyping

In three population-based cohorts of KoGES, a BMI above 25 and between 18.5 and 22.9 were considered obese and normal, respectively, in accordance with the Asia-Pacific guidelines of obesity classification system [38].

Hypertension was defined according to SBP ≥ 140 mmHg or DBP ≥ 90 mmHg. Individuals with SBP ≤ 120 mmHg and DBP ≤ 80 mmHg were allocated to the normotensive control group for comparison.

Dyslipidemia was defined by the presence of one or more of the following conditions: total cholesterol (TCHL) ≥ 240 mg/dL, low-density lipoprotein cholesterol (LDLC) ≥ 160 mg/dL, triglyceride (TG) ≥ 200 mg/dL, or high-density lipoprotein cholesterol (HDLC) < 40 mg/dL. In this study, LDLC was calculated using the Friedewald formula only when the triglyceride concentration was 400 mg/dL or less [39]. Individuals with TCHL < 200 mg/dL, LDLC ≤ 129 mg/dL, TG < 150 mg/dL, and HDLC ≥ 60 mg/dL were classified into a normolipidemic control group for comparison.

T2D was defined using the following criteria: fasting plasma glucose (GLU0) ≥ 126 mg/dL, plasma glucose 2 h after ingestion of 75 g oral glucose load (GLU120) ≥ 200 mg/dL, or glycosylated hemoglobin A1c (HbA1c) ≥ 6.5%. Individuals with GLU0 < 100 mg/dL, GLU120 < 140 mg/L, and HbA1c < 5.7% were classified as non-diabetic controls.

### 4.4. Quality Control across the Base and Target Data for PRS Derivation

The base data for PRS derivation were obtained from the summary statistics of GWA analyses for BMI using the KBA dataset of 57,110 individuals from the HEXA cohort. Associations between SNPs and BMI were tested by linear regression analysis adjusting for sex, age, and recruitment area using PLINK v1.07 (https://zzz.bwh.harvard.edu/plink/, accessed on 1 May 2023) [40]. The KBA genotype data of 13,595 individuals from KARE and CAVAS cohorts were used as the target data for computing the PRS in the present study.

The standard GWAS QC process removed SNPs with MAF < 1%, HWE *p* < 1 × 10^−6^, or imputation INFO score < 0.8 from both the base and the target datasets. In addition, SNPs with genotype missingness > 1% were further excluded from the initial target dataset. Ambiguous SNPs (i.e., those with complementary alleles, either C/G or A/T SNPs) across the datasets, duplicate SNPs, and SNPs on sex chromosomes were removed for subsequent PRS analysis. SNPs that were mismatched between the base and target data were not considered in the data QC because the base and target data were generated from the same genotyping platform. The BMI summary statistics of the base data were on the same genome build (Human GRCh37/hg19) as the target data.

Individuals with gender discrepancy or cryptic first-degree relatives were removed from the base and target datasets. In addition, individuals with genotype missingness > 1% or very high or low heterozygosity rates were further excluded from the initial target dataset. Finally, 6,916,878 variants for 57,110 individuals and 7,975,625 variants for 13,504 individuals remained in the base and target datasets, respectively (Figure 1). The detailed QC procedure for PRS analysis is presented elsewhere [41].

### 4.5. Derivation of PRS

PRSice-2 [26] software was used to derive the PRS from the QCed base and target data. As the target sample size was larger than 500 samples, the target file was used as the reference panel for the LD estimation in performing PRS calculation. For the LD clumping, r^2^ > 0.1 was applied. The phenotype data of BMI and the covariate data such as sex, age, and recruitment area from 13,504 individuals of the QCed target dataset were concomitantly incorporated in computing PRS. The best-fit PRS was selected for a given phenotype (here, BMI) at a *p*-value threshold where the model fit had the greatest *R*^2^ score. In calculating PRS using PRSice-2, the model fit was defined as [*R*^2^ of the Full model] − [*R*^2^ of the Null model], where the Full model was [*R*^2^ of BMI ~ PRS + SEX + AGE + AREA] and the Null model was [*R*^2^ of BMI ~ SEX + AGE + AREA].

### 4.6. Validation of PRS

To validate the best-fit PRS for BMI (PRS_BMI_), the association between PRS_BMI_ and BMI was tested by linear regression and Pearson’s correlation analyses. In addition, associations between PRS_BMI_ and obesity-related quantitative traits including blood pressure (SBP and DBP), glycemic traits (GLU0, GLU120, and HbA1c), and lipid traits (HDLC, LDLC, TG, and TCHL) were also tested by linear regression and Pearson’s correlation analyses. For these analyses, association *p*-values were obtained from linear regression with adjustment for age, sex, and recruitment area in the target dataset (about 13,000 subjects from KARE and RURAL cohorts). All quantitative traits except LDLC and TCHL were natural log-transformed before association analyses. The proportion of variance for the traits explained by the PRS was computed as the *R*^2^ obtained from a full model including both PRS and covariates (age, sex, and recruitment area) minus the *R*^2^ obtained from a model including covariates alone. In addition, the associations of PRS_BMI_ with obesity and related diseases such as T2D, hypertension, and dyslipidemia (including hyperglyceridemia, hyper-LDL cholesterolemia, and hypo-HDL cholesterolemia) were tested by logistic regression analyses adjusting for age, sex, and recruitment area in the target dataset. Statistical analyses for all association tests were performed using R software.

### 4.7. Assessment of PRS on the Prevalence and Incidence of Obesity-Related Diseases

The prevalence of obesity and related diseases was compared according to each decile group of PRS_BMI_ in about 13,000 subjects in the target data (from KARE and RURAL cohorts). The significance of the relationship between the disease prevalence and decile groups of PRS_BMI_ was measured by correlation and regression analyses using R software.

Kaplan–Meier survival analysis was used to assess the prognostic value of PRS on the incidence of obesity and related diseases in about 5400 subjects of the KARE longitudinal prospective cohort. In the Kaplan–Meier survival analysis, the incidence of obesity and related diseases over time was compared among quartiles of PRS_BMI_. The association between quartiles of PRS_BMI_ and disease incidence was further assessed by a log-rank test using R software (version 4.3.0).

## Figures and Tables

**Figure 1 ijms-24-11560-f001:**
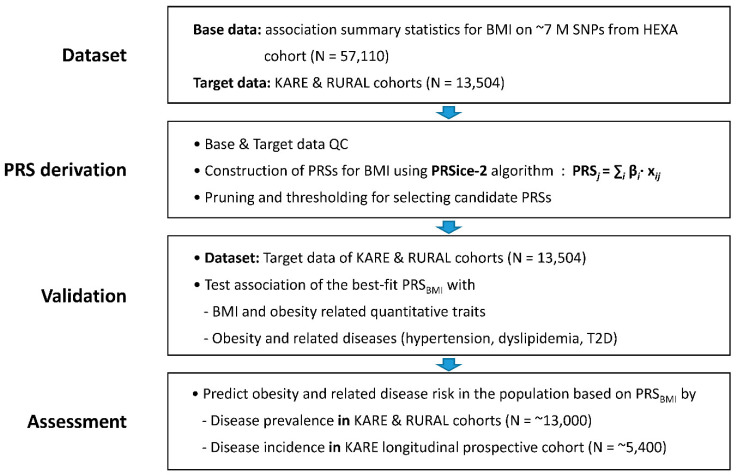
Study workflow demonstrating steps from data preparation to disease assessment.

**Figure 2 ijms-24-11560-f002:**
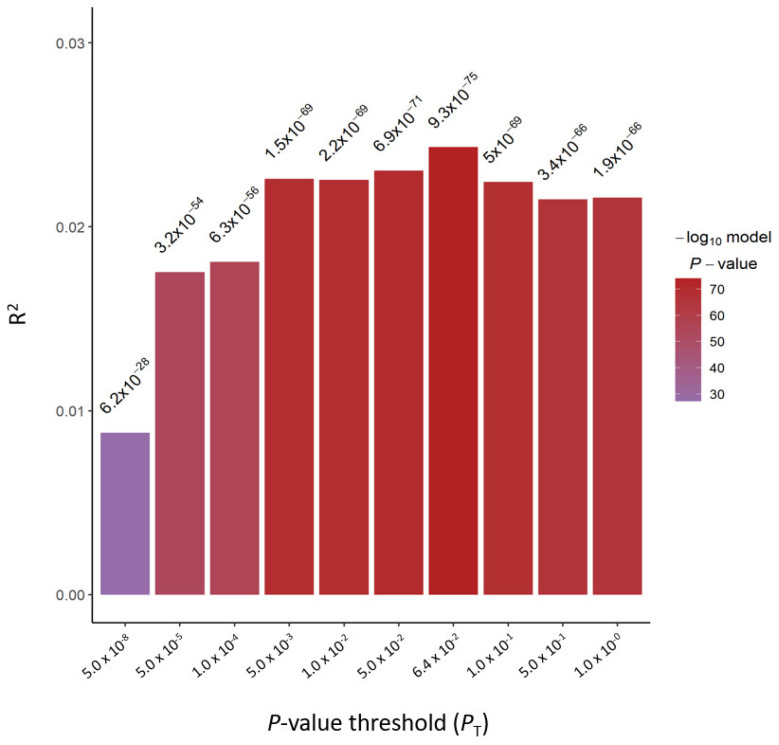
Bar plot showing the model fit of the PRS_BMI_ at *p*-value threshold. The *x*-axis indicates the *p*-value threshold (*P*_T_) used to select variants to be included in the PRS computation. The *y*-axis indicates PRS model fit (*R*^2^). The *p*-value of the logistic association test is plotted above the bar.

**Figure 3 ijms-24-11560-f003:**
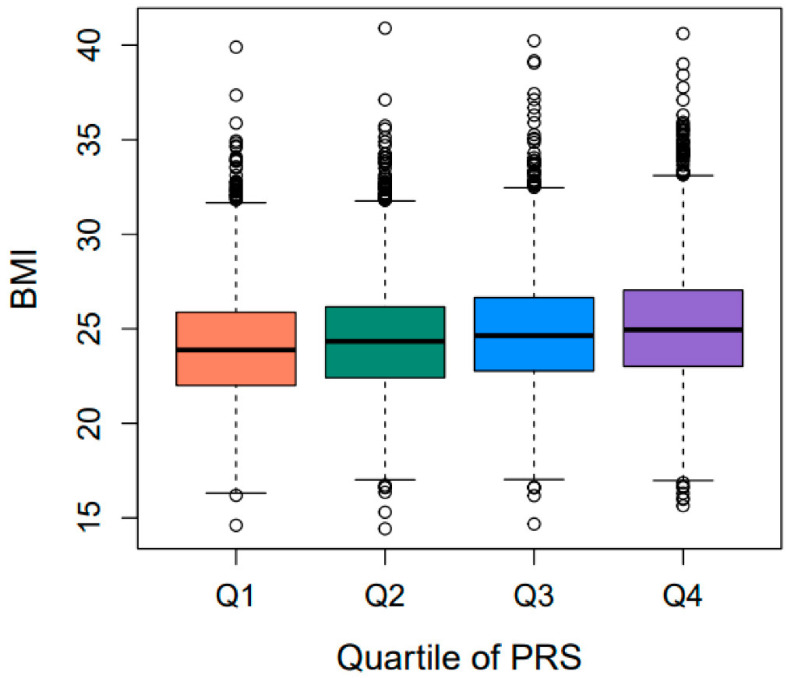
Comparison of BMI measurements among quartile groups of PRS_BMI_. Significant differences in BMI measurements between each quartile group were estimated by ANOVA.

**Figure 4 ijms-24-11560-f004:**
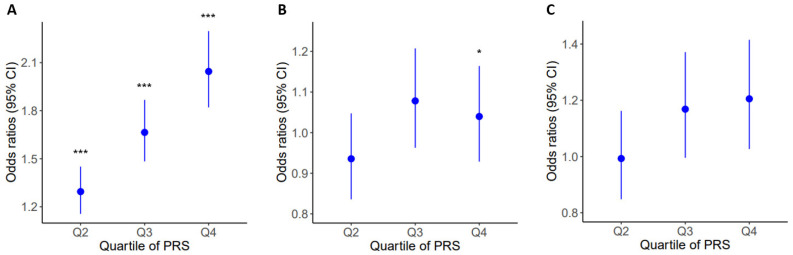
Relative risk of obesity (**A**), hypo-HDL cholesterolemia (**B**), and hypertension (**C**) for each quartile (Q2, Q3, and Q4) of the PRS_BMI_ compared with the lowest quartile (Q1) of genetic risk. Odds ratio of diseases between groups was assessed by Pearson’s chi-squared test (* *p* < 0.05, *** *p* < 0.001 vs. Q1).

**Figure 5 ijms-24-11560-f005:**
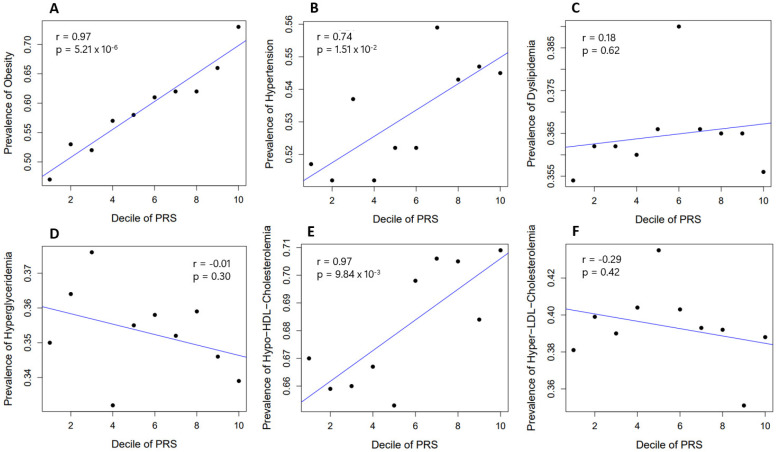
Comparison of the prevalence of obesity and related diseases by PRS_BMI_ decile. The significance of the relationship between disease prevalence and decile groups of PRS_BMI_ was measured by correlation and regression analyses. Subfigures (**A**–**F**) are for Obesity, Hypertension, Dyslipidemia, Hyperglyceridemia, Hypo-HDL-Cholesterolemia, and Hyper-LDL-Cholesterolemia, respectively.

**Figure 6 ijms-24-11560-f006:**
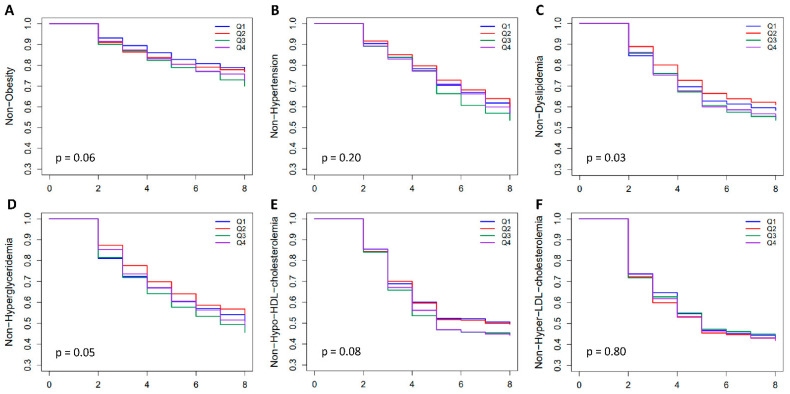
Comparison of the incidence of obesity and related diseases by PRS_BMI_ quartile. The prognostic value of PRS_BMI_ on disease incidence was assessed by Kaplan–Meier survival analysis in about 5400 subjects of the longitudinal prospective cohort. The *x*-axis indicates the follow-up stage. Follow-up data from 2001 to 2016 from the KARE cohort subjects were used in this analysis. Subfigures (**A**–**F**) are for Obesity, Hypertension, Dyslipidemia, Hyperglyceridemia, Hypo-HDL-Cholesterolemia, and Hyper-LDL-Cholesterolemia, respectively.

**Table 1 ijms-24-11560-t001:** Clinical characteristics of subjects in the study cohorts.

Variable	Base Dataset	Target Dataset	
	HEXA	RURAL	KARE
Female (%)	38,407 (65.4)	5010 (62.3)	2863 (52.4)
Age (year)	53.8 ± 8.0	58.5 ± 8.8	51.6 ± 8.5
BMI (kg/m^2^)	23.9 ± 2.9	24.5 ± 3.0	24.6 ± 3.0
SBP (mmHg)	122.5 ± 14.8	124.6 ± 17.4	120.9 ± 18.0
DBP (mmHg)	75.8 ± 9.7	78.6 ± 10.8	80.1 ± 11.2
FPG (mg/dL)	95.1 ± 19.8	98.1 ± 20.6	92.2 ± 21.2
OGTT120 (mg/dL)	NA	NA	132.4 ± 51.7
HbA1c (%)	5.7 ± 0.7	5.7 ± 0.8	5.8 ± 0.9
INS0 (μIU/mL)	NA	8.1 ± 4.8	7.5 ± 4.5
HDLC (mg/dL)	53.8 ± 13.2	45.1 ± 10.9	49.3 ± 11.5
LDLC (mg/dL)	119.3 ± 32.1	123.7 ± 31.6	120.5 ± 32.2
TG (mg/dL)	125.1 ± 85.6	146.2 ± 94.5	153.0 ± 110.4
TC (mg/dL)	197.4 ± 35.7	196.9 ± 35.3	199.0 ± 35.8

BMI, body mass index; FPG, fasting plasma glucose; OGTT120, oral glucose tolerance test 120 min; HbA1c, hemoglobin A1c; INS0, fasting insulin; TC, total cholesterol; TG, triglyceride; LDLC, low-density lipoprotein cholesterol; HDLC, high-density lipoprotein cholesterol; SBP, systolic blood pressure; DBP, diastolic blood pressure; NA, not available.

**Table 2 ijms-24-11560-t002:** Association of the best-fit PRS for BMI (PRS_BMI_) with obesity-related quantitative traits.

PRS	Related Disease	Trait	No of Samples	Correlation with QT	Linear Regression		
				Pearson r	β	SE	*p*
PRS_BMI_	Obesity	BMI	13,504	0.1590	0.0021	0.0001	1.36 × 10^−73^
	Hypertension	SBP	13,499	0.0289	0.0006	0.0001	2.63 × 10^−6^
		DBP	13,499	0.0299	0.0006	0.0001	2.90 × 10^−5^
	T2D	FPG	12,802	0.0029	0.0001	0.0001	4.39 × 10^−1^
		OGTT120	5192	−0.0104	−0.0002	0.0005	7.13 × 10^−1^
		HbA1c	6794	−0.0040	0.0000	0.0002	8.23 × 10^−1^
		INS0	5929	0.0490	0.0022	0.0008	8.15 × 10^−3^
	Dyslipedemia	HDLC	13,501	−0.0236	−0.0006	0.0002	9.07 × 10^−3^
		LDLC	13,177	−0.0026	−0.0001	0.0003	7.02 × 10^−1^
		TG	13,501	0.0078	0.0011	0.0005	2.50 × 10^−2^
		TC	13,501	−0.0062	−0.0001	0.0002	4.93 × 10^−1^

PRS_BMI_ is PRS calculated by linear regression analysis using BMI as a measure of obesity. All measurement traits except LDLC and TC were natural log-transformed and used for analysis. The proportion of variance for the traits explained by the PRS was computed as the R^2^ obtained from a full model including both PRS and covariates (age, sex, and recruitment area) minus the R^2^ obtained from a model including covariates alone. Abbreviations are as follows: PRS, polygenic risk score; QT, quantitative trait; BMI, body mass index; SBP, systolic blood pressure; DBP, diastolic blood pressure; T2D, type 2 diabetes; FPG, fasting plasma glucose; OGTT120, oral glucose tolerance test 120 min; HbA1c, glycosylated hemoglobin type A1c; INS0, fasting insulin; HDLC, high-density lipoprotein cholesterol; TG, triglyceride; TC, total cholesterol; LDLC, low-density lipoprotein cholesterol.

**Table 3 ijms-24-11560-t003:** Results of logistic regression analysis between PRS_BMI_ and obesity-related diseases.

PRS	Variable	N	OR	95% CI	*p*-Value
PRS_BMI_	Obesity	9671	1.031	1.027–1.035	8.73 × 10^−45^
	Hypertension	9757	1.008	1.003–1.013	6.84 × 10^−4^
	Type 2 Diabetes	3464	1.006	0.997–1.015	2.08 × 10^−1^
	Dyslipidemia	9283	1.001	0.996–1.006	6.52 × 10^−1^
	Hyperglyceridemia	13,501	1.002	0.998–1.006	3.15 × 10^−1^
	Hypo-HDL Cholesterolemia	5554	1.008	1.001–1.014	2.75 × 10^−2^
	Hyper-LDL Cholesterolemia	13,177	0.997	0.993–1.001	1.07 × 10^−1^

Analysis was performed on the target dataset. The same adjustments for age, sex, and recruitment area were performed. OR, odds ratio.

## Data Availability

Summary statistics of association analyses are available from the corresponding author upon reasonable request.

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
