# Peer review of "Development of a Polygenic Risk Score for BMI to Assess the Genetic Susceptibility to Obesity and Related Diseases in the Korean Population"

_ijms, 2023, doi:10.3390/ijms241411560_

Round 1
Reviewer 1 Report
The manuscript is very interesting, I have few suggestions
1) Enrich the introduction and provide more detail
2) Figure quality is not good, and even some wordings are not clear
3) Can be added some literature regarding leptin and other molecular literature
Minor revision needed
Author Response
The manuscript is very interesting, I have few suggestions
1) Enrich the introduction and provide more detail.
(Response 1)
According to the reviewer’s suggestion, we added more information to the introduction part to provide more detail. All modified parts are highlighted with yellow.
2) Figure quality is not good, and even some wordings are not clear.
(Response 2)
According to the reviewer’s suggestion, we improved the figure quality by increasing image ppi and font-size of axes/labels. We also revised some wordings in the figure legends to provide a clear understanding. All modified wordings in the figure legends are highlighted with yellow.
3) Can be added some literature regarding leptin and other molecular literature.
(Response 3)
According to the reviewer’s suggestion, we added some literatures regarding leptin and other molecular literature in the introduction part. All modified parts are highlighted with yellow.
Reviewer 2 Report
Our Esteemed Authorities
As you mentioned, the prevalence of obesity is increasing all over the world. Therefore, the frequency of obesity-related hypertension, diabetes, lipid metabolism disorders and cardiovascular diseases due to them is increasing.
Today, obesity and related diseases also determine a person's life comfort and life span. It is very important for everyone to be able to determine the criteria for determining or predicting obesity. In this beautifully designed and written study, it was found that the use of polygenic risk score according to BMI was beneficial in the chorea population.
The publication of this study in medical journals will be beneficial for both medical students, assistants, specialist physicians and professors. Expanding this study and conducting it in other regions and other communities will allow us to increase our knowledge on this subject and ultimately take precautions.
I would like to thank you, the authors, for this beautiful work and wish you success in your work.
Author Response
Our Esteemed Authorities
Point 1: As you mentioned, the prevalence of obesity is increasing all over the world. Therefore, the frequency of obesity-related hypertension, diabetes, lipid metabolism disorders and cardiovascular diseases due to them is increasing.
(Response 1)
Thanks for the reviewer’s comment.
Point 2: Today, obesity and related diseases also determine a person's life comfort and life span. It is very important for everyone to be able to determine the criteria for determining or predicting obesity. In this beautifully designed and written study, it was found that the use of polygenic risk score according to BMI was beneficial in the chorea population.
(Response 2)
Thanks for the reviewer’s comment.
Point 3: The publication of this study in medical journals will be beneficial for both medical students, assistants, specialist physicians and professors. Expanding this study and conducting it in other regions and other communities will allow us to increase our knowledge on this subject and ultimately take precautions.
I would like to thank you, the authors, for this beautiful work and wish you success in your work.
(Response 3)
Thanks for the reviewer’s comment.
Reviewer 3 Report
The authors developed new polygenic risk score for assesment of genetic susceptibility to obesity and related diseases in the Korean population. They used data obtained in previuos studies on large number of participants and the parameter that was derivated show strong correlation with body mass index (BMI).
The authors described statistical methods that they use in detail. Obtained results are presented with apropriate number of Figures and Tables. The quality of Figures 5 could be improved making y-axes titles more visable. There is not many literature data in this scientific field, so the originality of this manuscript is high. In discussion section, the authors emphasized the importance of the results of this study and thier polygenetic score for BMI.
The optional suggestion for authors is to add some literature data, if they exist, about SNPs variants frequences in other populations beside Korean as this information could increase interest to the readers and general significance of content.
Author Response
Point 1: The authors developed new polygenic risk score for assesment of genetic susceptibility to obesity and related diseases in the Korean population. They used data obtained in previuos studies on large number of participants and the parameter that was derivated show strong correlation with body mass index (BMI).
The authors described statistical methods that they use in detail. Obtained results are presented with apropriate number of Figures and Tables. The quality of Figures 5 could be improved making y-axes titles more visable. There is not many literature data in this scientific field, so the originality of this manuscript is high. In discussion section, the authors emphasized the importance of the results of this study and thier polygenetic score for BMI.
The optional suggestion for authors is to add some literature data, if they exist, about SNPs variants frequences in other populations beside Korean as this information could increase interest to the readers and general significance of content.
(Response 1)
Thanks for the reviewer’s comment. According to the reviewer’s comment, we improved the quality of figures by making axis titles and labels more visible. We also added the information of the minor allele frequency of each SNP in three ancestry groups (such as European, African, and East Asian) besides that in Korean in Supplementary Table 1. Minor allele frequency of each SNP in ancestry groups was obtained from 1000Genomes through dbSNP database (https://www.ncbi.nlm.nih.gov/snp/).